

# Temperature-size responses during ontogeny are independent of progenitors' thermal environments

Gerard Martínez-De León, Micha Fahrni and Madhav P. Thakur

Institute of Ecology and Evolution, University of Bern, Bern, Switzerland

## ABSTRACT

**Background:** Warming generally induces faster developmental and growth rates, resulting in smaller asymptotic sizes of adults in warmer environments (a pattern known as the temperature-size rule). However, whether temperature-size responses are affected across generations, especially when thermal environments differ from one generation to the next, is unclear. Here, we tested temperature-size responses at different ontogenetic stages and in two consecutive generations using two soil-living Collembola species from the family Isotomidae: *Folsomia candida* (asexual) and *Proisotoma minuta* (sexually reproducing).

**Methods:** We used individuals (progenitors; F0) from cultures maintained during several generations at 15 °C or 20 °C, and exposed their offspring in cohorts (F1) to various thermal environments (15 °C, 20 °C, 25 °C and 30 °C) during their ontogenetic development (from egg laying to first reproduction; *i.e.*, maturity). We measured development and size traits in the cohorts (egg diameter and body length at maturity), as well as the egg diameters of their progeny (F2). We predicted that temperature-size responses would be predominantly determined by within-generation plasticity, given the quick responsiveness of growth and developmental rates to changing thermal environments. However, we also expected that mismatches in thermal environments across generations would constrain temperature-size responses in offspring, possibly due to transgenerational plasticity.

**Results:** We found that temperature-size responses were generally weak in the two Collembola species, both for within- and transgenerational plasticity. However, egg and juvenile development were especially responsive at higher temperatures and were slightly affected by transgenerational plasticity. Interestingly, plastic responses among traits varied non-consistently in both Collembola species, with some traits showing plastic responses in one species but not in the other and vice versa. Therefore, our results do not support the view that the mode of reproduction can be used to explain the degree of phenotypic plasticity at the species level, at least between the two Collembola species used in our study. Our findings provide evidence for a general reset of temperature-size responses at the start of each generation and highlight the importance of measuring multiple traits across ontogenetic stages to fully understand species' thermal responses.

Corresponding author
Gerard Martínez-De León,
gerard.martinezdeleon@unibe.ch

## INTRODUCTION

Climate warming is causing widespread and rapid ecological impacts, which are likely to become more severe under intense warming scenarios (*IPCC, 2022*). These ecological impacts are diverse and might occur at various levels of biological organization, from organismal traits to ecosystems (*Harris et al., 2018*; *Harvey et al., 2022*). One of the key climate-driven ecological effects is body size reductions in warmer environments, particularly in ectotherms (*Gardner et al., 2011*; *Sheridan & Bickford, 2011*; *Verberk et al., 2021*). Given that body size is a fundamental trait influencing a myriad of key biological processes (*Brown et al., 2004*), such as metabolism (*Gillooly et al., 2001*) or predator-prey interactions (*Brose, 2010*), a reduction in body size due to warming can have important ecological consequences. For instance, smaller body sizes in warm environments help increase community persistence in consumer-resource systems (*Sentis, Binzer & Boukal, 2017*), whereas such reduction in body sizes of unicellular organisms can also decrease their productivity (*Malerba, White & Marshall, 2018*).

Body size reduction is considered a life-history response to warming (*Atkinson & Sibly, 1997*; *Kingsolver & Huey, 2008*), favoring earlier reproduction over growth in warmer environments (*Fryxell et al., 2020*; *Verberk et al., 2021*; *Wootton et al., 2022*). Alternatively, smaller body sizes could also be the result of a greater competitive advantage of smaller individuals under warming (*Ohlberger et al., 2011*; *Reuman, Holt & Yvon-Durocher, 2014*), caused in part due to greater metabolic costs in large individuals (*Malerba, White & Marshall, 2018*; *Riemer et al., 2018*; *Verberk et al., 2021*). Regardless of the ultimate cause, the proximate mechanism explaining smaller asymptotic body sizes at the individual level—commonly known as the temperature-size rule (*Atkinson, 1994*; *Ohlberger, 2013*)—is a warming-driven increase in growth or developmental rates during ontogeny (*Zuo et al., 2012*). This mechanism generally causes ectotherms to reach a given life stage, such as maturity, already at a smaller body size (*Forster, Hirst & Atkinson, 2012*; *Horne, Hirst & Atkinson, 2015*; *Verberk et al., 2021*). Significant variation in size-at-stage across thermal gradients can arise from varying thermal dependencies of growth and development (*Ohlberger, 2013*; *Verberk et al., 2021*). For instance, many organisms with indeterminate growth reach their asymptotic body size much later than their first reproduction (maturity) and, in these cases, size at maturity and asymptotic size are likely to display distinct responses to temperature (*Hoefnagel et al., 2018*; *Loisel, Isla & Daufresne, 2019*). As a consequence, the prediction of temperature-size relationships may not always be true across ectotherms. Yet, with a few exceptions (*e.g.*, temperature-size responding in opposite directions in uni- and multivoltine species; *Horne, Hirst & Atkinson, 2015*), the physiological and ecological mechanisms shaping variation in temperature-size responses remain poorly understood (*Verberk et al., 2021*).

Temperature-size responses during ontogeny may further be influenced by the thermal environment experienced by previous generations (*Verberk et al., 2021*), a phenomenon known as transgenerational plasticity (*Sgrò, Terblanche & Hoffmann, 2016*; *Bonduriansky, 2021*). For instance, eggs might be adaptively larger when laid at colder temperatures because they need better provisioning (*Liefting et al., 2010*), given that metabolic costs are

less sensitive to temperature (*i.e.*, more constant metabolism with changing temperatures) than total developmental costs (*i.e.*, longer development and therefore greater overall costs in colder environments; *Pettersen et al., 2019*). Egg provisioning can be viewed as a form of condition-transfer transgenerational plasticity, by which parents in favorable environments provide consistently higher offspring performance through better provisioning and/or inherited better condition (*Bonduriansky & Crean, 2018*). Temperature-dependent egg provisioning is highly relevant to understand variation in the size responses, because the thermal environment of the progenitors could induce carryover effects on the body size of following generations (*e.g.*, larger eggs developing into larger juveniles and adults; *Tully & Ferrière, 2008*; *Geister et al., 2009*; *Walczyńska et al., 2015*).

Transgenerational size plasticity could also arise independently of changes in egg size if other factors influencing the offspring phenotype are transmitted through the gametes (*e.g.*, methylation patterns affecting gene expression; *Bonduriansky & Crean, 2018*). Even though it is argued that quick acclimation of growth and developmental rates make the temperature-size rule a 'within-generation phenomenon' (*Forster & Hirst, 2012*), a transgenerational component may become more apparent when the thermal environments of progenitors and their progeny do not match (*Loisel, Isla & Daufresne, 2019*). One plausible prediction is that plasticity in body size could be constrained when the thermal conditions experienced by the parents differ greatly from the temperatures at which their offspring are exposed during development, as a result of the limits and costs of plasticity (*DeWitt, Sih & Wilson, 1998*; *Walczyńska, Kiełbasa & Sobczyk, 2016*; *Blanckenhorn et al., 2021*). This prediction implies the existence of anticipatory plasticity, which would reduce performance (*e.g.*, reduced growth, development or fecundity) when thermal environments change across generations (*Engqvist & Reinhold, 2016*; *Bonduriansky & Crean, 2018*). Alternatively, offspring raised in environments warmer than those of their progenitors could grow into smaller-than-expected adults compared to those raised in the same progenitors' environment ("plastic overshoot"), because of an inherited higher thermal sensitivity of developmental rates (*Loisel, Isla & Daufresne, 2019*). Indeed, deviations from the temperature-size rule may depend on the direction of the thermal change across generations (cold to warm *vs.* warm to cold; *Walczyńska et al., 2015*, *Loisel, Isla & Daufresne, 2019*). As a whole, understanding transgenerational effects of the temperature-size rule is particularly relevant to explain (the lack of) body size shifts when thermal environments vary across generations, such as in many multivoltine species in seasonal environments.

Here, we investigate the role of the thermal environment experienced by focal cohorts (F1) and their progenitors (F0) in determining temperature-size responses in two closely related Collembola species (from the same family, Isotomidae): *Folsomia candida* Willem 1902 and *Proisotoma minuta* Tullberg 1871. These collembolans are terrestrial invertebrates occurring in the litter and shallow soil layers of various temperate habitats (*e.g.*, heath, forests, farmland), often at high abundances (*Gisin, 1943*; *Fountain & Hopkin, 2005*). Despite the ecological similarities between the two Collembola species, they differ substantially in their mean body sizes (*F. candida*: 2,000–2,200 μm; *P. minuta*: 1,300–1,350 μm; details in Methods) and in their size plasticity when exposed to different

thermal environments (*Thakur et al., 2017*; *Marty et al., 2022*). In previous studies, average adult body size of *F. candida* (measured at the population level) was shown to decline in warmer environments, whereas that of *P. minuta* remained unaltered, which had implications for their ecological interactions (*Thakur et al., 2017*). Using the same model system, we aim to dive deeper into what underlies the variation in their thermal responses by investigating their plasticity at various ontogenetic stages, and exploring whether the progenitors' thermal environments modulate this potential plasticity. More specifically, we asked whether the thermal environments of the cohorts and their progenitors affect egg size and body size at maturity of the cohort, as well as egg size of their progeny (F2; Fig. 1). We predicted that (1) trait plasticity in different thermal environments would be affected by the thermal conditions experienced by the progenitors, constraining plasticity in the cohorts when their thermal environments and those of their progenitors do not match due to anticipatory transgenerational plasticity (*Walzer, Formayer & Tixier, 2020*). However, we expected that (2) trait plasticity would be affected to a greater extent by the thermal environment of the focal cohorts, given that growth and developmental rates would be more directly affected by the temperature experienced by the cohorts than those by their progenitors (*Forster & Hirst, 2012*). We also predicted that (3) body size at maturity would be much more responsive to warming than egg size (*Forster, Hirst & Atkinson, 2011*), because temperature-size responses tend to accumulate along ontogenetic growth (*Forster, Hirst & Atkinson, 2011*; *Horne et al., 2019*), and further because egg size may be subjected to trade-offs with clutch size due to limiting maternal investments (*Marty et al., 2022*). Given that previous findings have shown smaller asymptotic body- and egg sizes in warmer environments in *F. candida* but not in *P. minuta* (*Thakur et al., 2017*; *Marty et al., 2022*), we expected (4) smaller sizes in *F. candida* in warmer environments at a given ontogenetic stage (*e.g.*, egg size or size at maturity), while we expected that sizes in *P. minuta* would remain unaltered across temperatures. Additionally, given that temperature-size responses are strongly influenced by the thermal sensitivity of development, we measured thermal reaction norms of egg and juvenile development in the focal cohorts. We finally predicted that (5) development would be shortened in warmer environments in both Collembola species, likely being more thermally responsive than size-at-stage. This is because the size-at-stage is influenced by the coupling between growth and development (*e.g.*, no shifts in size-at-stage when both growth and development react in a similar manner to temperature changes), while development is more directly determined by temperature (*Birkemoe & Leinaas, 2000*).

## MATERIALS AND METHODS

### Experimental setup

Both species used in the experiment, *Folsomia candida* and *Proisotoma minuta*, were cultured for more than 1 year (encompassing 8–12 generations) in incubators set at either 15 °C or 20 °C under constant dark conditions and fed with dry yeast. Details of the animals that originated the cultures are provided in *Marty et al. (2022)*. The mode of reproduction is mainly asexual (parthenogenetic) in *F. candida* and sexual in *P. minuta*. In the case of *F. candida*—whose life-history is much better known given its common use

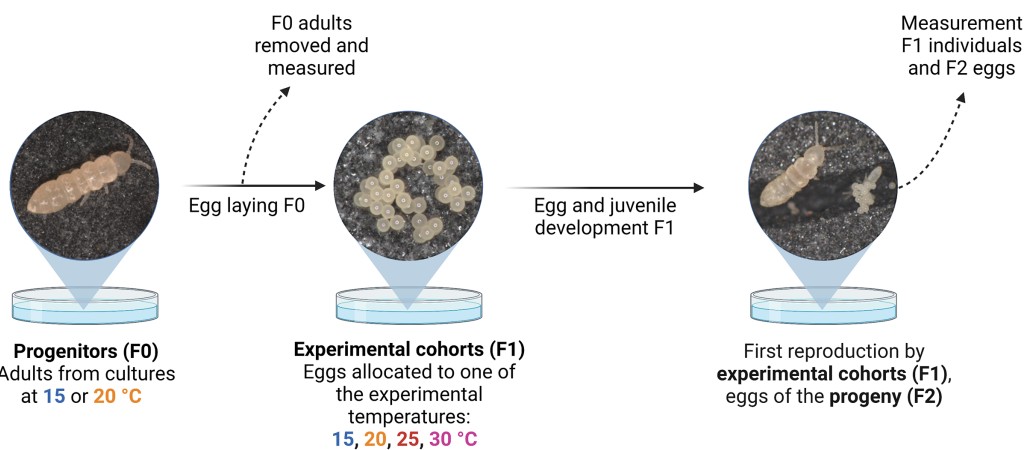

**Figure 1 Schematic representation of the experimental design.** Temperature-size responses of two Collembola species (*Folsomia candida* and *Proisotoma minuta*) were assessed in two consecutive generations (progenitors (F0) and experimental cohorts (F1)) using populations established in plates. Egg sizes of the progeny (F2) are regarded as a trait determined by the experimental cohorts (F1). Created with BioRender.com.

in ecotoxicological assays (*Fountain & Hopkin, 2005*)—lifetime reproductive output peaks at ~21 °C (*Mallard et al., 2020*), which can be regarded as an optimum temperature for the species.

We started the experiment by establishing populations of adult collembolans from several source cultures (*i.e.*, progenitors; F0) in order to obtain offspring of the same age for the following phases of the experiment (henceforth referred as experimental cohorts; F1). For this purpose, we added nine adults of *F. candida* or 11 adults of *P. minuta* into each of 80 Petri dishes (90-mm diameter) with a moist substrate of plaster of Paris and activated charcoal (9:1 mixture), in addition to dry yeast as food source provided *ad libitum* (Supplemental Methods). The plates with F0 adults were placed in their respective temperatures (*i.e.*, 15 °C or 20 °C; progenitors' temperature) and monitored every day until recently laid eggs were detected. As soon as we detected at least one clutch, all adults were removed from the plates, and their body length was measured (at 10X magnification). We found that the mean adult body length varied between the F0 individuals issued from the 15 °C and 20 °C cultures in *F. candida*, as individuals from 20 °C were 7.6% larger (mean ± SE; 2,063 ± 28 μm at 15 °C, $n = 175$; 2,220 ± 27 μm at 20 °C, $n = 178$; $P < 0.001$; Table S1), while mean adult body length did not differ in *P. minuta* (1,354 ± 26 μm at 15 °C, $n = 260$; 1,302 ± 26 μm at 20 °C, $n = 243$; $P = 0.158$; Table S1). The larger body sizes in *F. candida* at 20 °C could be the result of intraspecific competition (*Mallard et al., 2020*), given that animal densities were not controlled at this stage and could vary among cultures. Afterwards, we put the plates with fresh eggs belonging to the F1 cohorts in one of the four experimental temperatures (15 °C, 20 °C, 25 °C and 30 °C) (Fig. 1). We draw attention to the fact that all F1 eggs laid in the same plate (coming from the same F0 progenitors) were brought together to the same experimental thermal regime. The thermal range spanned the lowest culturing temperature (15 °C) and temperatures known to be deleterious for collembolan fecundity (30 °C; *Mallard et al., 2020*; *Martínez-De León et al., 2024*).

We established a total of 80 experimental units: 2 Collembola species × 2 progenitors' temperatures × 4 treatment temperatures × 5 replicates. A schematic summary of the experimental design is displayed in Fig. 1. We found that the 30 °C experimental treatment induced severe stress in both species, hindering their development: no eggs of *Folsomia candida* managed to hatch at this temperature, and only few eggs of *Proisotoma minuta* hatched but did not survive until maturity (Fig. S1). Consequently, we assessed the thermal reaction norms of size (*i.e.*, egg size and size at maturity) and development (*i.e.*, egg and juvenile development) of F1 individuals across three experimental temperature treatments (15 °C, 20 °C or 25 °C), originating from progenitors reared at two temperatures (15 °C or 20 °C).

To measure egg development time, we monitored F1 eggs on a daily basis, and noted the first day in which animals hatched at each Petri dish. When all individuals hatched, we reduced the total number of individuals below 50 (mean ± SD; *F. candida*: 42.2 ± 10.4 individuals; *P. minuta*: 25.1 ± 13.7 individuals; Fig. S2). This was done to control for potential effects of varying intraspecific competition on body size, caused when animal densities differ greatly among experimental units (*Mallard et al., 2020*). We kept monitoring the hatchlings every day until at least one individual reached maturity, that is, when new eggs were detected in the plate for the first time (*i.e.*, progeny from the experimental cohorts; F2). At this point, we collected and counted all collembolans from the plates, stored them in ethanol (70%), and measured their body length (at 10× magnification). No assessment of maturity at the individual level was possible since collembolans are ametabolous invertebrates with indeterminate growth, which means that there are no apparent morphological differences across life stages besides body sizes (*Stam, van de Leemkule & Ernsting, 1996*). Later, we took pictures of egg clutches (both from the experimental cohorts and their progeny) at 150X as soon as eggs were detected, which were later counted, and their diameter was assessed to estimate egg size. All size measurements were performed with KEYENCE VHX-970F (Keyence, Osaka, Japan). We assessed body size and development using individuals within cohorts instead of individually raised animals, as typically done in life-history experiments (*e.g.*, *Stam, van de Leemkule & Ernsting, 1996*). Our approach of using cohorts instead of individuals allowed us to understand the underlying mechanisms causing warming-driven body size reductions at the population level in Collembola, as reported in previous studies (*e.g.*, *Thakur et al., 2017*; *Robinson et al., 2018*; *Mallard et al., 2020*). In the present study, using cohorts removes the effect of size structure on mean population body sizes, and thereby allows us to focus on size-at-stage responses (*Ohlberger, 2013*). In addition, these collembolans thrive when raised in groups (*e.g.*, *Sengupta, Ergon & Leinaas, 2017*), particularly in the case of the sexually reproducing *P. minuta*. Nevertheless, individuals within cohorts are expected to display substantial variation in development and body sizes, and thereby yield greater uncertainty on the contribution of each individual on the estimated (population-level) trait values, compared to measurements made on isolated individuals.

## Statistical analyses

We modelled sizes (*i.e.*, egg diameter and body length at maturity) and development (*i.e.*, time spent in the transition between life stages) as a function of the progenitors' temperature (15 °C or 20 °C), the experimental temperature of the cohorts (15 °C, 20 °C, 25 °C or 30 °C) and Collembola species (*F. candida* or *P. minuta*). Given that trait plasticity in response to the experimental temperatures could be modulated by the progenitors' temperature and the species, we initially tested three-way interactions as progenitors' temperature × experimental temperature × species. Interaction terms and main effects were assessed *a priori* by means of data visualization (following *Zuur, Ieno & Elphick, 2010*), and selected or removed from the models based on AIC and their statistical significance. Final models obtained after simplifying the initial three-way interaction models are provided in Table 1. Given that size variables were measured at the individual level within cohorts, we employed linear mixed effects models (R package *lme4*, v.1.1-26; *Bates et al., 2015*) using experimental unit (*i.e.*, Petri dish containing a single cohort) as a random intercept to account for the dependency among individuals from the same cohort. Developmental variables were measured at the Petri dish level (*i.e.*, first day in which a life-stage transition was recorded), and were therefore analyzed using linear models. Furthermore, given that temperature-size responses are mainly revealed within the thermal range allowing for positive population growth (*Walczyńska, Kiełbasa & Sobczyk, 2016*; *Blanckenhorn et al., 2021*), we obtained a coarse estimate of the thermal performance curve of fecundity by modelling the mean egg production at the first reproductive event. In this case, we employed linear models using *per capita* egg production (*i.e.*, total number of eggs averaged by the number of adults in a plate) as a response variable. Using the same model selection approach as above, we additionally included mean body size at the Petri dish level as an additive term to account for potential effects of body size on fecundity at the first reproductive event (*Liefting et al., 2015*; *Tully, 2023*). However, one important limitation of our estimate of fecundity is that it contains information on both the number of F1 individuals that have reached maturity at the time of observation, as well as the number of eggs laid by each individual (actual fecundity), which warrants caution in the interpretation of these results. Linearity assumptions of all linear models were tested and visually inspected using the package DHARMa (*Hartig, 2022*).

Body length at maturity of the experimental cohorts was measured on the same age in all individuals of a given cohort, but it might be the case that only the largest individuals of the cohort reached maturity (*e.g.*, due to variation in the pace of development among the individuals of the cohort). In this case, the highest quantiles of the cohort's body size distribution are a better proxy of size at maturity than the mean cohort body size. We therefore tested changes in body size at the quantiles 0.75, 0.85, and 0.95 of the distribution using linear quantile mixed models (R package *lqmm*, v.1.5.6; *Geraci, 2014*), with experimental unit as a random intercept. To simplify the models and enable proper estimation of model parameters, we fitted separate models by species, using centered and scaled body size values. Testing interactions between progenitors' and experimental temperature precluded the adequate estimation of model parameters, so only main effects

**Table 1 Summary of the final models used to assess thermal reaction norms of body size and development.** Models were fitted as a function of the Collembola species, the progenitors' temperature experienced by the progenitors (F0) and the experimental temperature experienced by the cohorts (F1).

| Generation | Trait | Level of measurement | Final model |
|---|---|---|---|
| Progenitors (F0) | Adult body length | Individual | *Species × progenitors' temperature + (1 | plate)* |
| Experimental cohorts (F1) | Egg diameter | Individual | *Species + progenitors' temperature + (1 | plate)* |
| | Average body length at maturity | Individual | *Species × experimental temperature + (1 | plate)* |
| | Quantiles body length at maturity | Individual | *Experimental temperature + (1 | plate) - in both species* |
| | Egg development | Plate | *Species × progenitors' temperature × experimental temperature* |
| | Juvenile development | Plate | *Species + experimental temperature* |
| | Egg production at first reproductive event | Plate | *Species × experimental temperature* |
| | Egg diameter of progeny (F2) | Individual | *Species × experimental temperature + (1 | plate)* |

were tested in this particular analysis. Final models were selected based on AIC and statistical significance, similarly to the more conventional linear models described above. We performed *post-hoc* tests to obtain all *p*-values using the function *emmeans* (R package *emmeans*; v.1.7.0; Lenth, 2024). All statistical analyses were carried out in R statistical software (v.4.0.2; R Core Team, 2023).

# RESULTS

We found that the progenitors' temperature had overall minor effects on the traits expressed by the cohorts. Only egg development of *P. minuta* from their progenitors at 20 °C was shortened by 2.2 ± 0.35 days (predicted mean ± SE; Fig. 2; Table S2) at the experimental temperature of 15 °C. This reduction of egg development was not accompanied by changes in egg size (Fig. S3; Table S3). In contrast to the weak effects of progenitors' temperatures, experimental temperatures (*i.e.*, experienced throughout the development of the cohorts) strongly affected development-related traits in both Collembola species. Egg and juvenile development (Fig. 2) were faster in warmer environments, with a greater acceleration of development from 15 °C to 20 °C than from 20 °C to 25 °C in both Collembola species (Tables S2 and S4). Contrastingly, mean body size at maturity was less responsive to the experimental temperatures: size at maturity did not differ across experimental temperatures in *F. candida*, while in *P. minuta* size at maturity at 15 °C was slightly but significantly larger than at 20 °C and 25 °C (Fig. 3; Table S5). However, when assessing changes in the largest individuals of the distribution using quantile regression, we found that those of *F. candida* were smaller at 25 °C for quantiles 0.85 and 0.95, although this effect was only marginally significant for the 0.95 quantile (*P* = 0.051; Fig. 4A; Table S6). No such pattern was detected in *P. minuta*, as adult individuals at quantile 0.75 attained smaller sizes at 25 °C, but remained similar in size across experimental temperatures at higher quantiles (Fig. 4B; Table S7). Furthermore, egg size of the progeny (F2) was hardly affected by the experimental temperature: only eggs

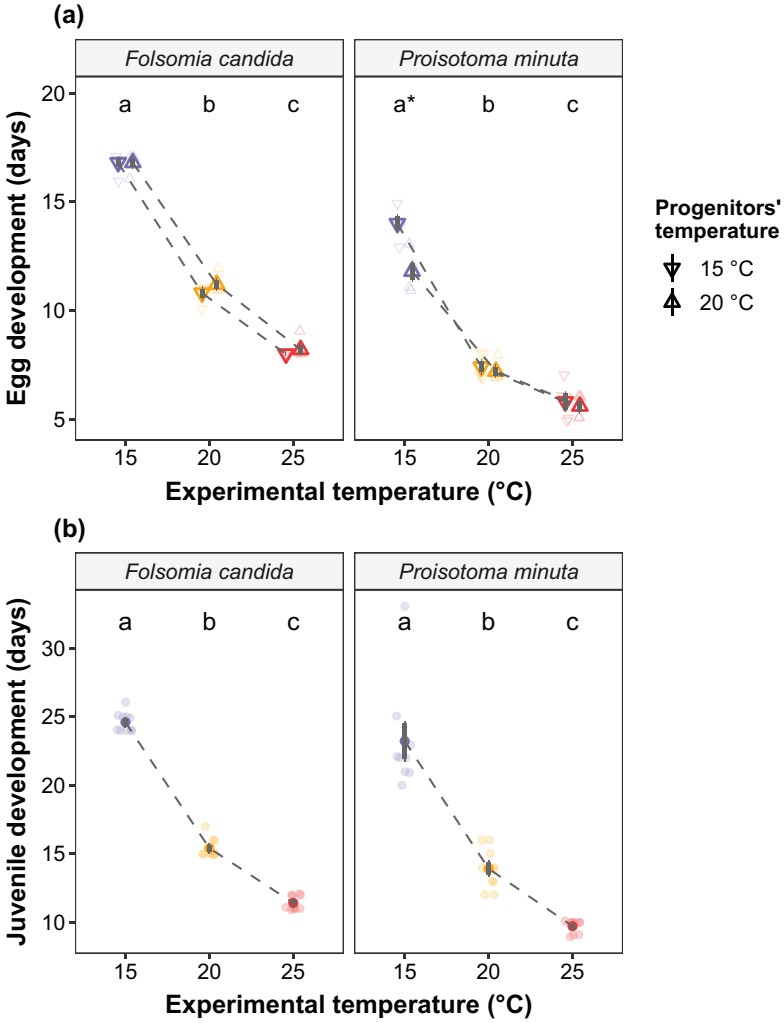

**Figure 2** **(A) Egg development and (B) juvenile development of the experimental cohorts (F1) as a function of the progenitors' and the experimental temperatures.** (A) Egg development is defined as the time from egg laying until first hatching. Shapes indicate different progenitor's temperatures: down-pointing triangle ∇, 15 °C; up-pointing triangle Δ, 20 °C. Solid points represent means, dark bars represent standard errors, and faded points are raw data. Stars show significant differences between progenitors' temperatures: *$P < 0.05$. Different lowercase letters indicate significant differences ($P < 0.05$) between experimental temperatures within Collembola species. For the full model output, see Table S2. (B) Juvenile development is defined as the time from first hatching until first reproduction (maturity). Solid points represent means, dark bars represent standard errors, and faded points are raw data. Different letters indicate significant differences ($P < 0.05$) between experimental temperatures within Collembola species. The variable "progenitors' temperature" was not retained in the final models (Table 1), and is therefore not displayed in the figure. For the full model output, see Table S4.

laid by *P. minuta* at 25 °C were larger (+6.8%) than those at 20 °C, but not different to those laid at 15 °C (Fig. 5; Table S8). Finally, mean egg production at the first reproductive event of F1 was strongly affected by the experimental temperature in *Folsomia candida*, with greater egg production at 25 °C, but not in *Proisotoma minuta* (Fig. S4; Table S9). Mean body size of F1 individuals did not influence their egg production and was therefore not included in the final models (Table 1).

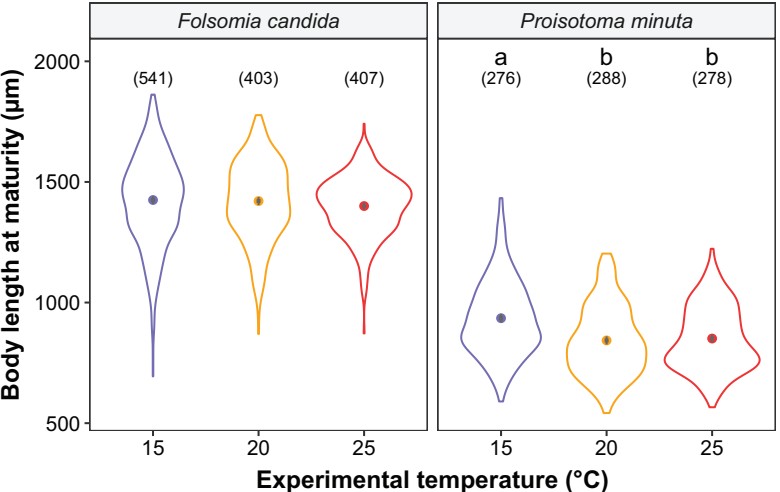

**Figure 3** **Body length at maturity (*i.e.*, when the first reproductive event of the cohort was detected) of the experimental cohorts (F1) as a function of experimental temperature.** Violin plots represent body length distributions, solid points show means, and grey bars are standard errors. The number of individuals measured in each treatment is displayed in brackets. Different lowercase letters indicate significant differences in mean body length ($P < 0.05$) between experimental temperatures within Collembola species. For the full model output, see Table S5.

## DISCUSSION

In this study, we tested how different thermal environments experienced by cohorts (F1) and their progenitors (F0) affected size and development at different ontogenetic stages in two Collembola species. We found that trait plasticity in the cohorts was weakly modulated by the thermal environments of their progenitors, while the temperature conditions experienced by the cohorts were largely driving plasticity in their traits, particularly development, in both species. Yet, we also found that trait plasticity in response to the temperature regimes varied non-consistently between the two collembolans (*e.g.*, *P. minuta* had larger egg sizes at 25 °C, while size-at-maturity of the largest individuals of the cohorts declined in *F. candida* at 25 °C), pointing out to distinct mechanisms shaping plasticity among the traits measured in our study. We acknowledge that using cohorts instead of individuals may have precluded the detection of more apparent patterns of trait plasticity, while the reduced number of progenitor (15 °C and 20 °C) and experimental (15 °C, 20 °C, 25 °C and 30 °C) temperature regimes could also have masked some of the expected trait plasticity, especially since these collembolans likely experience a wider range of thermal conditions in the wild (*Lembrechts et al., 2022*). Hence, even if our findings regarding temperature-size responses across generations should be interpreted with caution, they still offer valuable insights into how temperature can affect trait plasticity within populations.

### Transgenerational plasticity did not constrain temperature-size responses

Our findings generally indicate that the progenitors' thermal environments did not modulate plasticity in body size and development in the following generation. This is

**(a)** *Folsomia candida*

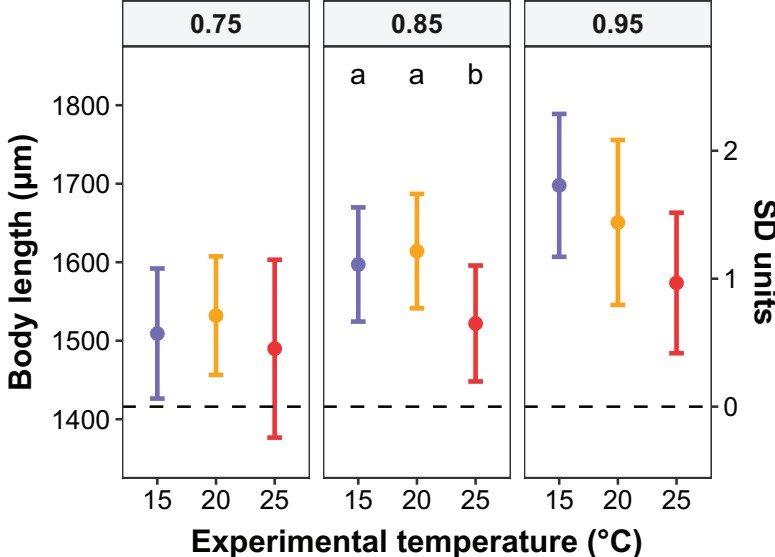

**(b)** *Proisotoma minuta*

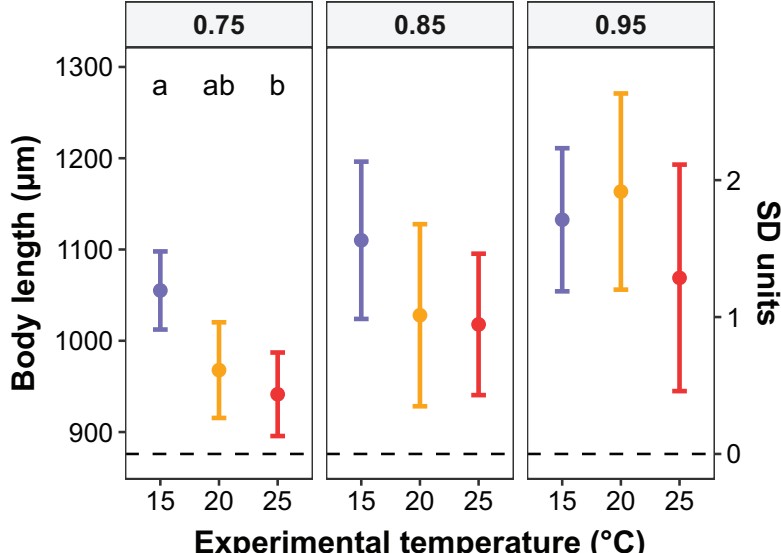

**Figure 4** **Changes in the body size distribution at maturity of F1 individuals at quantiles 0.75, 0.85 and 0.95 as a function of experimental temperature in (A)** *Folsomia candida* **and (B)** *Proisotoma minuta*. Raw and scaled body lengths (measured in SD units) are displayed on the left and the right side of the y-axis, respectively. Body lengths were scaled (mean = 0; standard deviation = 1) to allow the estimation of parameters in the linear quantile mixed models. Different lowercase letters indicate significant differences ($P < 0.05$) between experimental temperatures within each quantile. For the full model outputs, see Tables S6 and S7.               

consistent with the notion that temperature-size responses are reset at the start of each generation (*Forster, Hirst & Atkinson, 2011*; *Forster & Hirst, 2012*), which means that temperature-driven transgenerational plasticity may have minor importance in determining size-at-stage and development in the two Collembola species examined (but

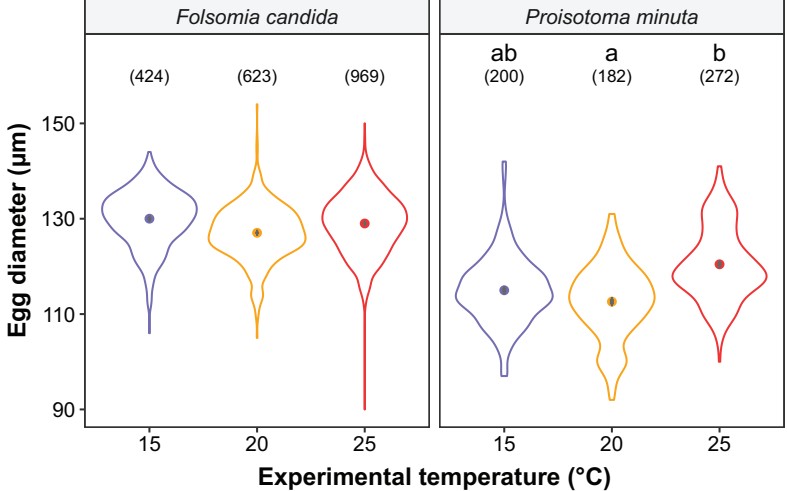

**Figure 5 Egg diameter of the progeny (F2) as a function of experimental temperature.** Violin plots represent egg diameter distributions, solid points show means, and grey bars are standard errors. The number of eggs measured in each treatment is displayed in brackets. Different lowercase letters indicate significant differences in mean egg diameter ($P < 0.05$) between experimental temperatures within Collembola species. For the full model output, see Table S6.

see *Walczyńska et al., 2015*). However, we found that traits concerning the embryonic phase (egg development in F1 and egg size in F2) were, in some cases, affected by the temperature experienced by the previous generation. We found that egg development of the cohorts (F1) was ~16% shorter at 15 °C in *P. minuta* when eggs were laid by progenitors in warmer environments (20 °C compared to 15 °C). These eggs performed better (*i.e.*, shorter development) despite the mismatch between the thermal environments of progenitors and their offspring, which would show support for transgenerational plasticity based on condition-transfer effects (*Bonduriansky & Crean, 2018*; *Bonduriansky, 2021*). Interestingly, the effect of the progenitors' temperature on egg development faded out when eggs were developed at higher temperatures and were not even detected in *F. candida* for any of the treatment combinations. Several factors (*e.g.*, egg provisioning or epigenetic factors in the gametes) could be responsible for this variation in transgenerational plasticity and, therefore, provide additional support for the occurrence of condition-transfer effects (*Bonduriansky & Crean, 2018*). In this regard, we show that egg size in F1, a proxy of egg provisioning (*Geister et al., 2009*), did not differ between progenitors' temperatures, which implies that other mechanisms must be involved if condition-transfer effects were to explain our findings. Even though the eggs laid at the higher temperature may have started to develop more quickly before they were transferred to the colder environment (*e.g.*, *Geister et al., 2009*), such an experimental artifact is unlikely to fully explain our findings. In particular, the magnitude of developmental shortening (−2.20 days) was much greater than the time lag from oviposition to the experimental transfer of eggs (few hours to less than a day). Condition-transfer effects in the context of the temperature-size rule were also suggested by *Walzer, Formayer & Tixier (2020)*, who showed that predatory mites (*Amblydromalus limonicus*) developed

consistently faster when their progenitors were exposed to heat wave conditions. In addition, they reported larger sizes at maturity (following a reverse temperature-size pattern) when offspring matched the thermal environments of their progenitors, indicating the occurrence of anticipatory transgenerational plasticity (*Walzer, Formayer & Tixier, 2020*).

In the case of egg size in F2, we show that eggs laid at higher temperatures (25 °C) were ~7% larger in *P. minuta*. We speculate that this is likely the result of greater maternal provisioning of eggs at 25 °C (*Pettersen et al., 2019*). Developmental costs increase greatly beyond a certain high temperature threshold, so mothers should increase egg sizes to provide enough energy to their offspring to fuel the higher energetic costs (*Pettersen et al., 2019*). However, why this is the case for *P. minuta* but not for *F. candida*, where egg sizes remained similar across temperatures, is unclear. This difference is especially intriguing considering the known plasticity of egg sizes in the latter species (*Stam, van de Leemkule & Ernsting, 1996*; *Tully & Ferrière, 2008*; *Marty et al., 2022*). As opposed to our results, *Liefting et al. (2010)* reported smaller egg sizes at 20 °C than at 16 °C in the collembolan *Orchesella cincta*, conforming to the temperature-size rule. Likewise, previous work using the same study system revealed that egg sizes declined from 15 °C to 20 °C in *F. candida* due to the existence of trade-offs with clutch sizes in the cooler environment (*i.e.*, larger eggs in smaller clutches), while no trade-offs were detected in warmer conditions (*Marty et al., 2022*). Such inconsistency between these findings and the present results (*i.e.*, no change in egg sizes from 15 °C to 20 °C) might be explained by the trade-offs between egg size and clutch size, which were not accounted for in the present study. Alternatively, we speculate that differences in egg sizes between 15 °C and 20 °C in *Marty et al. (2022)* might be a transient response due to the shorter acclimation time of collembolan cultures (~6 months), compared to the present study (~1 year). Overall, we showed that egg sizes mostly remained unaltered across different thermal environments, confirming our expectation that eggs typically display weak temperature-size responses.

## Responses of body size at maturity affected by growth and development

We found that development generally differed strongly at various thermal regimes in both Collembola species, whereas body size at maturity responded more weakly or not at all. Size at maturity is ultimately determined by both developmental and growth rates, so weak responses of size at maturity suggest a strong coupling between both rates across temperatures in these collembolans. *Sengupta, Ergon & Leinaas (2017)* reported similar thermal reaction norms of development and growth in *Folsomia quadrioculata*, resulting in a lack of response of size at maturity across temperatures. In contrast, asymptotic body sizes (*i.e.*, maximum body size in organisms with indeterminate growth) declined at warmer temperatures following the temperature-size rule, as also shown by *Mallard et al. (2020)* in *Folsomia candida*. Altogether, these results indicate that size at maturity and asymptotic size might be shaped by different mechanisms, supporting the idea that no singular mechanistic explanation exists for the temperature-size rule across ontogeny (*Hoefnagel et al., 2018*; *Verberk et al., 2021*). Given that size at maturity and asymptotic size

can display different responses to temperature (*Loisel, Isla & Daufresne, 2019*), we suspect it to explain the discrepancy between the findings by *Thakur et al. (2017)* and the results from this study, namely the difference in temperature-induced (adult) body size responses. In the previous study, mean adult body sizes were substantially smaller in warmer environments in *F. candida* but remained unaltered in *P. minuta* (*Thakur et al., 2017*). Given that size measurements were done on adults from populations established during several weeks (*i.e.*, much longer than age at maturity), it is likely that a large proportion of the measured adults were close to their asymptotic body sizes, reaching values more similar to the asymptotic body sizes reported by *Mallard et al. (2020)* than to the sizes at maturity shown in the present study.

In addition, our results obtained with quantile regressions showed that individuals of *F. candida* belonging to the highest quantiles (*i.e.*, largest individuals of the cohorts) attained smaller sizes in warmer environments, whereas in *Proisotoma minuta* a similar response was found for lower quantiles and average body sizes. While it is unclear why temperature affected distinct aspects of the body size distribution of the cohorts between these species, we suspect that differences in the strength of intraspecific competition between species might explain this inconsistency. Strong intraspecific competition has been reported in *F. candida*, in which few competitively superior individuals monopolize resources through interference (*Le Bourlot, Tully & Claessen, 2014*), thereby reducing mean body growth rates at the population level (*Mallard et al., 2020*). Such interference competition could induce great variation in the progress of development within cohorts, causing more competitive individuals to grow quicker and reach size at maturity earlier than the rest of the individuals in the cohort (*Le Bourlot, Tully & Claessen, 2014*). This would make body sizes in larger competitive individuals to be more regulated by temperature, while growth in smaller individuals from the cohorts would be more affected by resource acquisition (*Mallard et al., 2020*). Remarkably, *Stam, van de Leemkule & Ernsting (1996)* reported that individuals of *F. candida* developed in isolation reached smaller sizes at maturity in warmer environments, matching our results from the highest quantiles of the cohorts. As opposed to *F. candida*, evidence on the strength and the type of intraspecific competition in *P. minuta* is still lacking. We speculate that weaker interference competition in *P. minuta* could induce more synchronous development within cohorts (*Le Bourlot, Tully & Claessen, 2014*), which could explain why lower quantiles of the size distribution (*e.g.*, quantile 0.75) and average body sizes were more responsive to warming. Furthermore, the range of body sizes at maturity in the cohorts was narrower in *P. minuta* (mean range across experimental temperatures ± SD: 720.3 ± 106.3 µm) than in *F. candida* (981.3 ± 162.8 µm; see Fig. 3), which could provide some support for the hypothesis of more synchronous development within our cohorts of *P. minuta*.

## Thermal reaction norms differed among traits and between species

Even though the underlying mechanisms driving trait plasticity can differ across ontogenetic stages, it is plausible that similar patterns might have emerged in the two closely related species. However, this was not the case in the species that we used in this

study, confirming previous work showing that these species had distinct body size responses to temperature (*Thakur et al., 2017*; *Marty et al., 2022*). Here, we extended on these studies by measuring development and size-related traits in cohorts exposed to various thermal environments. It has been previously showed that *Folsomia candida* consistently displayed greater plasticity in warmer environments than *Proisotoma minuta* for adult sizes (*Thakur et al., 2017*) and egg sizes (*Marty et al., 2022*). An explanation based on the mode of reproduction, which differs between *F. candida* (asexual) and *P. minuta* (sexual), could possibly clarify the distinct degree of plasticity displayed in both species. Species typically reproducing by parthenogenesis (asexual reproduction) may rely heavily on phenotypic plasticity to cope with environmental changes, whereas sexually reproducing species could display less plasticity and instead rely more on genetic variation (*Ponge, 2020*). In the context of the temperature-size rule, it follows that parthenogenetic species would display stronger size responses than sexually reproducing species. Yet, in the present study we found that the sexually reproducing *P. minuta* showed greater plasticity than *F. candida* in some traits (*e.g.*, egg size in F2), supporting the view that phenotypic plasticity needs to be assessed simultaneously for several traits given that plasticity of one trait might be associated with robustness in another trait (*Liefting et al., 2015*). Therefore, the idea that the degree of plasticity can be generally characterized at the species level, based on the mode of reproduction, seems unlikely according to our results. We also note that considerable genetic variation can evolve in parthenogenetic species in the form of distinct lineages (as in the case of *F. candida*), and therefore display substantial variation in trait values at the species level (*e.g.*, asymptotic body size, egg size; *Tully & Ferrière, 2008*; *Mallard, Farina & Tully, 2015*).

Another aspect of our between-species comparison is that other traits varying between *F. candida* and *P. minuta* could additionally affect their body and egg size responses, for instance the differences in their body sizes (smaller terrestrial species -in our case, *P. minuta*- are more likely to reduce their body size in warmer environments; *Horne, Hirst & Atkinson, 2015*) or their thermal performance curves of fitness metrics (*e.g.*, population growth rates; *Walczyńska, Kiełbasa & Sobczyk, 2016*; *Blanckenhorn et al., 2021*). Nevertheless, our estimate of fecundity (a major component determining population growth) did not change across thermal environments in *P. minuta*, while it increased strongly from 15 °C to 25 °C in *F. candida*. This suggests that the lack of size responses in *F. candida* cannot be explained by experimental temperatures being beyond its "optimal thermal range" (*Walczyńska, Kiełbasa & Sobczyk, 2016*). These findings are expected to be robust despite the limitations associated with our estimate of fecundity (*i.e.*, combining information on both the number of individuals reaching maturity and their reproductive output), and are further supported by previous studies assessing thermal reaction norms of fecundity in *F. candida* (*Mallard et al., 2020*).

## CONCLUSIONS

Given their limited abilities to track suitable thermal environments in space, litter and soil-dwelling collembolans may be particularly exposed to changes in their thermal conditions induced by climate warming. Our findings help us understand how

temperature influences trait plasticity in these soil invertebrates and potentially in other similar terrestrial ectotherms. We found that temperature-size responses are mostly driven by the thermal environments experienced at a given generation, providing support for the negligible carryover effects of temperature on body size across generations, at least for the range of temperatures examined in our study. Finally, based on our findings, we propose that ecological studies investigating population and community responses to warming can have a stronger mechanistic basis by better understanding plasticity in multiple traits measured along ontogeny and across species.

## ACKNOWLEDGEMENTS

We are grateful to Catherine Peichel, Gerald Heckel and Claudia Bank for several stimulating discussions that inspired this study, as well as Aleksandra Walczyńska and Thomas Tully for reviewing our manuscript and providing many constructive suggestions. We also thank our student assistants and Ludovico Formenti for maintaining the collembolan cultures.

### Funding

Madhav P. Thakur received support from the Swiss State Secretariat for Education, Research and lnnovation (SERI) under contract number M822.00029 and from the Swiss National Science Foundation (grant number: 310030_212550). The funders had no role in study design, data collection and analysis, decision to publish, or preparation of the manuscript.

### Grant Disclosures

The following grant information was disclosed by the authors:
Swiss State Secretariat for Education, Research and lnnovation (SERI): M822.00029.
Swiss National Science Foundation: 310030_212550.

### Competing Interests

The authors declare that they have no competing interests.

### Author Contributions

- Gerard Martínez-De León conceived and designed the experiments, analyzed the data, prepared figures and/or tables, authored or reviewed drafts of the article, and approved the final draft.
- Micha Fahrni performed the experiments, authored or reviewed drafts of the article, and approved the final draft.
- Madhav P. Thakur conceived and designed the experiments, analyzed the data, authored or reviewed drafts of the article, and approved the final draft.

### Data Availability

The complete dataset and R script used in this study are available at Figshare: Martínez-De León, Gerard; Fahrni, Micha; Thakur, Madhav (2024). Martinez-De Leon et al. 2024 Peer J-TSR generations ontogeny-Dataset. figshare. Dataset. https://doi.org/10.6084/m9.figshare.23042231.v1.

The complementary results and figures as well as model outputs are available in the Supplemental File.

## Supplemental Information

Supplemental information for this article can be found online at http://dx.doi.org/10.7717/peerj.17432#supplemental-information.

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
