# Peer review of "Temperature-size responses during ontogeny are independent of progenitors’ thermal environments"

_PeerJ, doi:10.7717/peerj.17432_

## Round 0.1 · original submission · Minor Revisions

Dear Dr. Martínez-De León,

After this review round, both reviewers provided very positive decisions regarding your manuscript. Both reviewers indicated the need for minor reviews, emphasizing introduction and discussion improvements. Still, you will see that the required changes may need more time than expected for a minor review. Therefore, I will grant you a major review period for you to be able to deliver the revised version.

Sincerely,
Daniel Silva

·

Basic reporting

The manuscript meets all the criteria for a professional scientific article in terms of the form, structure, language and literature referencing.

Experimental design

The research questions are well defined and timely.

I have some comments about the experimental design, but these are more about clarifying an experimental approach than about the criticisms.

The tests are rigorous and ethically correct.

The methods are generally well described, although I lack information on the two species studied.

Validity of the findings

I have some concerns about the interpretation of the results, but I give more details in the next section.

Additional comments

I have previously reviewed this manuscript for another journal. I appreciate the improvements the authors have made in the current version of the text.

My main concern was about the true sample size, which should refer to the 9 and 11 females from which the experimental cultures were initiated. In the current text version, the authors have decided to refer to their study as a cohort. This approach changes the perspective, but it is still not known what the inclusion of each of the F0 females in further experimental cultures was. Are the authors sure that all 9/11 females laid eggs, the eggs hatched and the more or less similar number of offspring per female were used in further analyses?

Another concern I have relates to the differences between the two species studied. On lines 111-112 of the current text it says "Despite some similarity between the two Collembola species, their thermal responses are shown to be different". Please clarify what is similar and what are the differences between these species. Such information could greatly aid the interpretation of the data. In addition, I have previously asked whether there is a difference in the standard body size of these species and I did not find an answer to this question in this version of the text. In this respect, I think an introductory paragraph describing the species in more detail in the Methods section could be helpful.

Some more detailed comments are the following:

l. 62-63 – “As a consequence, temperature-size relationships may not always be true in ectotherms.” – I do not understand this statement.

l. 156-158 – was the difference in size between F0 females of F. candida at two temperatures significant?

l. 162-164 – 1. What method was used to ensure that F1 from each F0 mother was more or less equally distributed between the experimental thermal regimes? 2. What was a sample size within a replicate? 3. I would like to point out something very important. The direction of temperature change was always the same for individuals from 15°C, but it was different for those from 20°C. The direction of temperature change was found to be crucial in another paper cited here by the authors (Walczyńska et al. 2015). Consequently, to avoid this effect for this study, the correct comparison of progenitors’ vs. ontogeny effect should be to compare F1 from 15 °C and 20 °C for F0 15 °C females with F1 from 20 °C and 25 °C for F0 20 °C females. Of course, the authors may have other solution, but I think that this issue should be solved.

Results and Discussion – Proper interpretation of the results is not possible without information on the optimum temperature for each species.

Fig. 3 – why is there no distinction between progenitors’ temperature?

·

Basic reporting

The article is well written, the English seems to me to be of high quality, the references cited are on the whole relevant, the introduction is on the whole clear and the elements discussed in the discussion help us to better understand and take a step back from the results presented.
The figures are neat, although I would like to make a few suggestions to improve their coherence and readability.

Experimental design

The experimental protocol is well presented and justified. However, I think that the paper would benefit from clarifying in the introduction the fact that the asymptotic size is not identical to the size at maturity in springtails and that therefore the predictions that we could make about the effects of temperature on the adult body size (=asymptotic size) do not apply as simply to the size at maturity. This point has been raised in the discussion, but it would be worth mentioning it clearly in the introduction, leaving the predictions that could be made about the effects of temperature on size at maturity more open.
The way in which the data has been analyzed and the results presented and discussed seems to be relevant.

Validity of the findings

The results presented are generally well supported by relevant statistical analyzes and clear figures.

The measurement of fecundity and the way in which it is interpreted and discussed are somewhat debatable. As the observations were made every day on cohorts of around fifty springtails, it is impossible to know how many springtails may have participated in the production of the first clutches observed.
This measure of fecundity is therefore likely to be very variable depending on the number of springtails which were able to lay eggs during the last observation interval and the fecundity analyzed which is the number of eggs / number of females is not a very reliable measure of fecundity at maturity because it integrates the probability that an individual lays eggs at this age, a value that we have no way of estimating. It is therefore probably a fairly biased proxy for fecundity at maturity. It's interesting to show this data but I think clarifications would help readers take a step back from what is shown or not, and not let people believe that it is a direct measure of fecundity at maturity. In a nutshell, adding a few sentences in the methods, results and discussion could help clarify this point.

The analysis of the quantiles of body length could maybe done if possible on untransformed variable and presented on the same scale as for figure 3 to help understand and compare the results.

Additional comments

--- Abstract ----
2 “Warming generally induces faster developmental and growth rates, producing smaller individuals in warmer environments at a given ontogenetic stage (a pattern known as the temperature-size rule)” -> I do not think that TSR is expected to apply for any ontogenetic stage. Given that growth is expected to slow down at low temperature, it is expected that juveniles of the same age will be smaller when raised at lower temperature than the one raised at warmer temperature. It is only on the long term that the growth trajectories are expected to cross and individuals raised at cold temperature manage to reach larger sizes. So to avoid any confusion maybe rephrase with something like that: ‘Warming generally induces faster developmental and growth rates and individuals that have developed in warmer environments usually reach a smaller adult asymptotic size than the ones who have grown in colder conditions (a pattern known as the temperature-size rule).’

12 “during their entire ontogenetic development (i.e., from egg laying to maturity)” -> I am not very comfortable with the “ontogenetic development” expression in this manuscript given that the development of springtails does not stop at sexual maturation since they continue to grow significantly afterwards. Could you rather write “exposed their offspring in cohorts (F1) from egg laying to first egg laying (maturation) to various thermal environments (15 °C, 20 °C, 25 °C and 30 °C)”. It may be worth verifying throughout the manuscript that there is no such ambiguity regarding “ontongenetic development”. At least remove the “entire” and replace it with the first part of the ontogenetic development or explain that when you refer to ontogenetic development here you put aside the part of the development which follows the first laying.

15 that that -> that

56 explaining smaller body sizes -> explaining smaller adult asymptotic body sizes

58-68
-> this is maybe the place where you could add a few words regarding the distinction between asymptotic body length and size at maturity in species with indeterminate growth like the Collembola… I think that it's important to make it clear in the introduction that the asymptotic size of adults, which generally decreases as temperature increases, is not the same thing as the size at sexual maturity: in a number of species, growth continues after sexual maturation. The predictions that can be made about the effects of temperature on asymptotic size do not therefore necessarily apply directly to size at maturity. In my opinion, this distinction is not sufficiently clear at this stage. It is probably useful to introduce the idea that some species can continue to grow after they have started to reproduce.

96 -> could you explicit what you mean by smaller than expected. Could you explain what your prediction is based on?

106 you could cite Willem 1902 and Tullberg 1971

132 -> To my knowledge, the two articles cited in support of the prediction (4) do not talk about the size of springtails at maturity. So the prediction about size at maturity does not seem to be well supported. On the other hand, the authors could mention the articles that have documented the effects of temperature on the asymptotic size of springtails. If one wishes to extrapolate these observations to potential effects of temperature on maturity, it should be clearly stated that this remains largely speculative. Given the speculative nature of the effects of temperature on maturity, it is also possible to indicate that we may be interested in the existence of such effects without having any particular predictions about their nature.

136 -> I would suggest to remove “considerably”

138 -> Could you clarify what do you mean by “the coupling between growth and development”?

--- Methods ---

234 -> I am not sur to understand why it was necessary to center and scaled the body length given that separate models have been done for each species. The disadvantage of doing this is that you lose the value of the length measurements in the results provided.

The measurement of duration and the precision of these measurements (duration of egg development) need to be explained. Given that the exact moment of egg laying or hatching cannot be observed, it is more a question of an estimated duration with an interval for the start and end of the event. Could you explain how was this taken into account in the analyses?

177.
“The progenitors’ temperature affected adult body sizes in F0 individuals” -> given that it is probably an effect of density rather than a direct effect of temperature, it is maybe better to rephrase in a more neutral way. Something like “we found that the mean adult body length varied between the individuals issued from the 15 and 20°C cultures.”

156
(15 °C, 20 °C, 25 °C, 30 °C) -> (15, 20, 25 and 30°C) ?

164-> I would remove the “with an increasing thermal gradient”. It’s confusing and unnecessary.

173 “assessed the thermal reaction norms of body size (i.e., egg size and size at maturity) and” -> could be rephrased as : "assessed the thermal reaction norms of egg size and egg development time and size and age at first reproduction (maturity) of F1 individuals... "

176 We monitored the complete -> To measure the egg development time, we monitored...

178 Given the potential importance of the density in your experiments, it could be worth giving the range of the density (or adding a figure in the supplementary to show the distribution of densities). Maybe also add a word or two to explain why you used different densities for the two species and if you measured the number of females and males in the Pm cohorts, especially if you estimate the number of eggs produced per female.

190 -> Eggs generally increase slightly in volume in the hours following their laying. Is this phenomenon likely to have influenced the measurements?

196 -> you could also cite Mallard 2020.

197 -> You could also mention the limits linked to this experimental choice.

197 -> Figure S7 from Mallard 2020 could be used to argue that in your designed experiment the density of springtails are low enough to have relatively small density dependence effects. You could try to compare your density to the Collembola density that we measured in our experiments.

202 "We modeled body size (i.e., egg diameter and body length at maturity)"-> I find it questionable to equate the diameter of an egg just after laying with a measurement of body size. -> We modeled egg diameter and body length at maturity..."

218 219. Is it possible to clarify this sentence?

224. You could also mention Tully 2023, where the figure S1 illustrates the relation between Collembola length and clutch size.
https://www.frontiersin.org/articles/10.3389/fevo.2023.1112045/full
It is also may be the place to mention the fact that only a portion (unmeasured) of the individuals in each cohort participated in the production of the eggs that were counted.

234 "proper estimation" -> explain why. I do not understand the need to scale and center the body size values especially given that analysis is done on each species independently. I think that it would be much easier to present the analysis on the untransformed variable (I am sure the results will be the same). Figure 4 could be fused to figure 3 and presented with the same units which would enable making visual comparison.

248 you could also refer to Figure S2 with Table S3.

251 It would be nice to have Fig.S3 grouped in a single four-panel figure with Fig 2 so that one can easily catch the parallel effects of temperature on the two traits.

255 significantly -> slightly but significantly larger

258 I personally would have appreciated seeing the effects of temperature on the different quantiles on the same figure to be able to easily compare them. To do this, it would be necessary to bring together the data from figures 3 and 4 on the same figure. In my opinion, the scale in SD units is unnecessarily confusing..

264 "fecundity at first reproductive event of F1". As mentionned ealier I do not think that you can reasonably claim to have measured "fecundity at first reproductive event". Please rephrase with something like "mean egg production at first reproductive event".

-- Discussion --
275 maybe mention the traits that varied non-consistently.

276 "pointing out to distinct mechanisms shaping plasticity among the traits measured in our study." -> I am not convinced by this claim.

278 Explain why?

279 You could also say that temperature range 15-20° is relatively narrow compared to what the Collembola may endure in the wild.

286. I would be less affirmative on this point given that the range of temperatures studied remains relatively narrow

288 are reset -> "are largely reset ... at least for the traits examined here."

289 warming-driven -> temperature-driven

290 has generally minor importance -> may have relatively minor importance ... in out two Collembola.

293 "experienced by the previous generation" -> Perhaps you could add that this is quite logical because it is difficult to see how the size of a laid egg could be determined by temperature after laying...

294 15.7 -> ~16%

310 with a daily followup, on average the eggs have spent 12h at 20° and the developmental rate is twice as fast at 20° as at 15°C (Figure 2), thus 12h at 20°=24h at 15°C...

318, 326, 338 the stability of the egg size in F candida that you observe reminds me the stability of egg size during lifespan that I report in Tully 2023 (Figure 8 and 9). -> canalized trait?

408 come at the expense -> be associated with?

415 A major caveat -> Another aspect of..

417 You could explain what you mean by absolute body size?

418 With the oxygen limitation hypothesis you would rather expect Pm to be less likely to reduce its size in warmer environments.

424 you could also cite Mallard 2020, Figure 2 shows that the temperatures that you have used are clearly within the range of optimal temperatures.

432 lack of -> negligible ... at least for the temperatures investigated here.

---

## Round 0.2 · Minor Revisions

Dear Dr. Martínez-De León,

After this new review round, I believe yourmanuscript is almost accepted for publication in PeerJ. One of the authors recommended the acceptance of the manuscript, while the other one indicated the needs for minor reviews.

Sincerely,
Daniel Silva

·

Basic reporting

The manuscript meets the scientific standards in all aspects.

Experimental design

There is one, but fundamental, doubt that I have, which I refer to below. The other methods are correctly applied.

Validity of the findings

The validity of the results depends very much on this crucial point I mentioned above.

Additional comments

I appreciate the improvements the authors have made to the text. I generally agree with the way my previous concerns have been addressed.

However, there is one point that I need to emphasise because, frankly, I do not know what to do with this issue.

The authors have responded honestly and in detail to my concern about the contribution of F0 individuals to the F1 generation, which is the one studied in the study. However, the authors' response made me even more sceptical. It seems that there is a high possibility that the F1 generation for each species is derived from only one mother (at least such a possibility cannot be excluded). I still see this as a major concern, because in principle such offspring from one mother should be treated as replications, while the true sample size is 1. This is because of the generally strong maternal effect, which cannot be controlled for in this study anyway. I am sorry to have to say this, especially as all the rest of the methodology has been applied correctly and the results are very clearly presented. However, this concern does make me doubt the validity of the results.

However, I accept that I may be wrong as I am not as familiar with collembolans as a study organism.

My solution is to leave the decision to the editor based on the response of the other reviewer.

I recommend "minor revisions" because there is no option for my case.

·

Basic reporting

no comment

Experimental design

no comment

Validity of the findings

no comment

Additional comments

The authors have taken care to provide precise and convincing answers to the various points raised. They took full advantage of the reviews to improve and clarify their article. I found the exchanges constructive, and I would like to thank the authors for their attentiveness to the suggestions that were made. The manuscript in its current form deserves to be published. (Congratulations :-)

For the caption of FigS1, maybe change "solid" by "large" and "faded" by "small" or modify the plot so as to have the same color transparency as in figure S4.

---

## Round 0.3 · accepted · Accept

Dear Dr. Martínez-De León,

I am pleased to inform you that your manuscript has been formally accepted for publication in PeerJ.

Sincerely,
Daniel Silva

·

Basic reporting

No comments.

Experimental design

No comments.

Validity of the findings

No comments.

Additional comments

I am grateful to the authors for this clarification and accept the argument. I am also quite relieved because, in general, I share the second reviewer's enthusiasm for the way in which the manuscript is prepared and presented.

I recommend publication of this manuscript in its present form.